# Verbenone Affects the Behavior of Insect Predators and Other Saproxylic Beetles Differently: Trials Using Pheromone-Baited Bark Beetle Traps

**DOI:** 10.3390/insects15040260

**Published:** 2024-04-09

**Authors:** Matteo Bracalini, Guido Tellini Florenzano, Tiziana Panzavolta

**Affiliations:** Department of Agriculture, Food, Environment and Forestry, University of Florence, 50145 Florence, Italy; tellini@dream.coop

**Keywords:** attraction, anti-aggregation, *Ips sexdentatus*, *Orthotomicus erosus*, *Thanasimus formicarius*, *Temnochila caerulea*

## Abstract

**Simple Summary:**

Our study evaluated the impact of verbenone, a common bark beetle aggregation inhibitor, on several saproxylic beetles in a pine forest in Tuscany (Italy). Verbenone pouches were added to traps baited with bark beetle pheromones, and we compared the captures from these traps with those of pheromone-only traps in spring–summer 2023. The captures contained 9440 beetles from 32 families and 57 species, with 80% being bark beetles. Predator beetles, beneficial insects which naturally control pest populations, made up 17%, including species not previously studied in combination with verbenone. While captures of some species were reduced in pheromone plus verbenone traps (including some pests), others showed no change, and some increased (including a few predators). Verbenone notably increased saproxylic beetle diversity when it was present in traps, implying potential implications of its use in limiting the aggregation of bark beetle pests on susceptible trees.

**Abstract:**

In our study, we assessed the effects of verbenone, the most widely studied bark beetle aggregation inhibitor, on saproxylic beetles in a Mediterranean pine forest in Tuscany. Verbenone pouches were devised in the laboratory and then applied to *Ips sexdentatus* pheromone traps so that their catches could be compared to those of traps containing just the pheromone. The trial was carried out in spring–summer 2023, and insect catches were collected every two weeks. A total of 9440 beetles were collected that belonged to 32 different families and 57 species. About 80% of the captures were bark beetles, mainly *Orthotomicus erosus*. Beetle predators accounted for about 17% of the captures, with a total of 12 species. Some of these predator species had not yet been studied in relation to verbenone effects, like other saproxylic beetles recorded in this study. A significant reduction in captures was recorded for some beetles (e.g., *I. sexdentatus* and *O. erosus*), while for other species, no differences emerged, and in some cases, captures increased significantly when verbenone was present in the traps (i.e., *Hylurgus ligniperda*, *Corticeus pini*, and *Aulonium ruficorne*). The diversity of caught saproxylic beetles increased significantly in the verbenone traps, highlighting possible implications of the use of verbenone when managing bark beetle outbreaks.

## 1. Introduction

In forest ecology, it is well understood that the assembly of saproxylic beetles varies depending on the condition of the host wood. Initially, more demanding beetle species colonize the tree, followed by later saproxylic species in the subsequent stages of wood decay [1]. The first type of saproxylic beetles mainly comprise bark beetles, which belong to the subfamily Scolytinae (Coleoptera: Curculionidae) [2]. Among bark beetles, there are species that attack live hosts (primary bark beetles) and species that exploit dead or severely stressed trees (secondary bark beetles) [3]. Species that arrive when trees are alive or early after death are more aggressive as they have to face host defenses. For this reason, they may also display gregarious behavior as adults [4], which is crucial to achieving a mass attack on the host tree, such as in some *Dendroctonus* and *Ips* species [5].

This behavior requires intraspecific communication through volatile substances, such as aggregation pheromones, that attract conspecifics on the same tree to overcome host defenses [3]. However, an excessive density of bark beetles on the host tree can lead to intraspecific competition. To avoid intraspecific competition, another signal is required, something which indicates that the host is no longer suitable, such as the monoterpene ketone verbenone [4]. It is not clear whether bark beetles can control the amount and production of verbenone as it seems to mainly be produced by microorganisms associated with bark beetles rather than by them directly; however, among those studied, most bark beetles are inhibited by this substance [4].

This is true mainly for primary bark beetles; strongly secondary wood-boring species may be indifferent to or attracted by verbenone [4]. Indeed, if this substance is an indicator of host tissue quality, it could thus inhibit early successional insects, such as bark beetles, that require fresh host tissue [6]. In fact, this substance is heavily produced in more degraded tissue; in addition, it is associated with an advanced bark beetle attack which greatly impacts wood decomposition, preparing the habitat for later saproxylic species [1,7]. For all these reasons, verbenone may be an exploitable signal for late-successional beetles that require a later stage of wood decay.

After decades of applied research, verbenone became widely known as the main bark beetle aggregation inhibitor for aggressive species. Several studies tested its efficacy in protecting trees from the attacks of the most damaging bark beetles, showing promising results [8,9,10]. This led to the production of several commercial formulations which are now registered in North America for the protection of individual trees or forested areas. These products entail a variety of verbenone dosages, used alone or in combination with other non-host volatiles, as well as different methods of dispersal ranging from polyethylene pouches to flowable emulsions that are applied to tree boles (i.e., SPLAT^®^) [11,12,13].

Considering the potential use of verbenone in forest protection, understanding the effect of this substance on the saproxylic beetle community in more detail would be useful for avoiding possible side effects. In this study, the effect of verbenone on the attractance of an aggregation bark beetle pheromone was tested using the saproxylic beetle community of a Mediterranean pine forest. More precisely, a commercial blend of the aggregation pheromone for *Ips sexdentatus* (Boern) was taken into consideration. This species can act as a primary bark beetle, occasionally becoming a pest in Mediterranean pine forests, leading to the application of phytosanitary control measures.

## 2. Materials and Methods

The present investigation was conducted in 2023 within a coastal pine stand spanning approximately 70 hectares, situated between Marina di Grosseto and Principina a Mare in the Province of Grosseto, Italy (42°42′17.22″ N, 10°59′47.59″ E). This privately owned forest plot primarily consisted of *Pinus pinaster* Aiton, with a minor presence of stone pine (*Pinus pinea* L.). From 2014 onward, the increased availability of pine trees weakened by the maritime pine bast scale *Matsucoccus feytaudi* (Ducasse) led to a notable rise in the population density of *I. sexdentatus*. Furthermore, the bark beetle *Orthotomicus erosus* (Wollaston) was also favored, even on pines already infested with *I. sexdentatus* or the other major pine bark beetle, *Tomicus destruens* (Wollaston), as well as on those trees highly stressed by abiotic factors [14]. Starting in 2016, phytosanitary measures, such as the prompt cutting of infested trees and mass trapping, were carried out, negatively affecting the population density of *I. sexdentatus*, which was reduced to sustainable levels in 2022 [14]. However, trappings were planned again in 2023 to continue monitoring both *I. sexdentatus* and *O. erosus*.

On 1 March 2023, a total of 10 “Super Forest” bark beetle slot traps (Serbios s.r.l., Badia Polesine, Rovigo, Italy) were positioned along a transect placed on the southwestern borders of the forest plot, with an approximate spacing of 150 m between each trap. The positioning of the transect did not vary from the original one planned in 2016 for the first phytosanitary measures. However, due to extensive felling of infested pines over the years, in 2023, eight traps ended up in open areas among the seedlings of a new plantation of stone pines. The remaining two traps were in similar conditions as in 2016, with much more cover resulting from the remnants of adult maritime pines. The traps were baited with a commercial blend of the *I. sexdentatus* aggregation pheromone, SuperWood (Serbios S.r.l.). This pheromone is known to also be effective toward *O. erosus*, and it contains ipsenol, ipsdienol, and 2-methyl-3-buten-2-ol as its main components. No pesticides were used to treat the collection containers. The collection of catches occurred every 14 days from March to October with pheromone renewal every 28 days, approximately 20 days earlier than the dispensers’ expected lifespan. Deviating from the manufacturer’s instructions (50 days) was necessary since dispensers deplete faster in Tuscany’s coastal pine stands due to the warm climate [15].

To mitigate the possible side effects of mass trapping, such as the capture of non-target beetles, slot traps, which are already size-selective, were modified as follows: a 6 mm mesh screen was added to the top of the collection container, following the methodology of Martìn et al. [16]; additionally, three 60 mm × 8 mm escape windows were provided immediately above the mesh screen (one central window on one side and two lateral windows on the other side), as per the approach of Bracalini et al. [14]. These customized traps inhibited the capture of endangered non-target species like *Chalcophora detrita* (Klug), as well as other large beetles that are present in the study area.

To assess the effect of verbenone on various saproxylic beetle species, polyethylene dispensers were developed in the laboratory and applied to five of the ten traps within the study area. A 50 μm thick polyethylene pouch, designed to provide a 40 mg/24 h evaporation rate at room temperature (22 °C), was selected. The substance, (1S)-(−)-verbenone (≥93% purity) (Merck Life Science S.r.l.; Milano, Italy), was introduced to the pouch using a wetted pad of folded absorbent paper (90 cm^2^). The field-applied dosage was 3 mL/dispenser, ensuring a lifespan of approximately two months, although renewal occurred every 14 days to counteract verbenone degradation. Overall, we used 5 traps baited only with the pheromone (Ph-traps) and 5 traps baited with pheromone in combination with verbenone (Ph + V-traps). To avoid the effect of trap position, Ph-traps were alternated with Ph + V-traps along the transect, switching their positions every two weeks. This bi-weekly verbenone rotation continued until the study’s conclusion, ensuring a balanced and consistent distribution of repellent treatment across all traps.

All captured beetles, in addition to the targeted *I. sexdentatus* and *O. erosus*, were systematically collected and examined in the laboratory. Species were identified either by using taxonomic keys or relying on the counsel of experts. For the analysis of verbenone’s effects, our attention was concentrated on beetles as they constitute one-quarter of the entire dead-wood community, including plants and fungi [1]. Furthermore, among insects, this order has both the highest number of saproxylic taxa and related wood microhabitats [17]. Captures were divided into four categories: bark beetles, insect predators, other saproxylic beetles, and non-saproxylic beetles. The observations of a 2015 study showed how omitting Staphylinidae does not affect the results of studies on saproxylic biodiversity; thus, this beetle family was not considered here because they are known to be a complex taxonomical group constantly in revision; consequently, its species are not easily determined [18].

### Statistical Analysis

In addition to the effect of verbenone treatment, our sampling was influenced by two important sources of bias: the seasonal effect and the trap location effect. They can both heavily influence the number of beetles caught. To take them into account, owing to the characteristics of the involved variables, we used a GAMM (Generalized Additive Mixed Model) analysis, a method that can include in the model one or more additive variables (each of which is treated as a smoother), as well as random effects [19,20]. In our case, the smoother pertained to the seasonal effect, using the Julian date as a variable, and the random effect was the trap location. The dependent variable (no. of beetles/trap/date) was assumed to follow a Poisson distribution. For statistical analyses, we used the “mgcv” package [21,22] within an “R” environment [23]. Furthermore, captures were also compared by calculating the following indices of diversity [24]:
Simpon’s diversity index:
D = Σ[(ni/N)^2^]
2.Shannon index (H′):
H′ = −Σ[ni/N × log(ni/N)]
where ni is the number of individuals of one particular species found, and N is the total number of individuals found.
3.Pielou’s evenness:
E = H′/ln S
where S is the total number of species.

## 3. Results

Over the whole study period, from 1 March 2023 to 24 October 2023, a total of 9440 beetles were collected in the 10 plot traps used (Table 1). When possible, specimens were identified at the species level, but in several cases, they were identified at the genus or family level. In fact, as soon as they were captured, the insects could damage each other, especially if live predators were present inside the traps. The absence of an insecticide inside traps increases this kind of problem, leading to a higher percentage of specimens whose identification must be carried out based only on some of their body parts. The catches contained beetles belonging to 32 different families, accounting for 57 different species.

Most captures were bark beetles (79.74%), followed by predators (17.24%), while the other saproxylic beetles represented only 1.84% of the total; finally, non-saproxylic beetles comprised 1.17%. Ten species of bark beetles were captured; however, the high number caught (7528) was almost entirely due to *O. erosus*, which represented 94.33% of bark beetle captures. This was followed by *Hylurgus ligniperda* (Fabricius) (3.61%) and *I. sexdentatus* (1.51%); the other seven species were trapped only sporadically. As regards predators, 12 species belonging to eight different families were trapped (Table 1). The most frequent was the Trogossitidae *Temnochila caerulea* (Olivier), which accounted for more than half of the predator captures (59.95%), followed by the Monotomidae *Rhizophagus depressus* (Fabricius) (13.14%), the Zopheridae *Aulonium ruficorne* (Olivier) (10.13%), the Tenebrionidae *Corticeus pini* (Panzer) (5.84%), the Cleridae *Thanasimus formicarius* (Linnaeus) (5.71%), and, finally, the Histeridae *Plegaderus otti* Marseul (4.12%). The other six species, belonging to the Carabidae, Elateridae, and Histeridae families, were captured only sporadically. Finally, 23 species (in 16 families) of other saproxylic beetles and 12 species (in 8 families) of non-saproxylic beetles were trapped during the whole study period. As previously stated, Staphilinidae were not considered, though 27 specimens belonging to this family were trapped.

Comparing captures from the Ph-traps with those from the Ph + V-traps, some significant differences emerged. Firstly, as expected, the seasonal effect on captures was very high as it was statistically significant for almost all species (Table 2). However, despite this, an effect of verbenone also emerged. Even though *I. sexdentatus* was present at a low population density (only 114 captures and a low number of attacked trees), a significant difference emerged as the Ph-traps caught more *I. sexdentatus* specimens than the Ph + V-traps (Table 2 and Figure 1). Overall, the presence of verbenone reduced captures by 67.44%. However, some traps caught fewer beetles when lured only with Ph, and this was particularly true when captures were very low (Figure 2). Verbenone caused an even higher reduction in captures of *O. erosus* (76.20%). All traps caught significantly more *O. erosus* specimens when only the pheromone was present, and this was true for all sampling dates (Table 2, Figure 1b and Figure 2b). Regardless, two of the used traps, those under pine cover, caught most of the individuals. Among the bark beetles tested, only catches of *H. ligniperda* significantly increased (by 49.72%) when verbenone was added (Table 2).

Regarding predators, verbenone had different effects depending on species. Verbenone significantly reduced (by 72.60%) *T. formicarius* catches when it was added to the pheromone trap (Table 2). On the contrary, catches of *C. pini*, *P. otti*, and *A. ruficorne* increased significantly by 73.33%, 51.11%, and 77.78%, respectively. However, Figure 3 and Figure 4 show that these species were caught mainly on a few sampling dates and in a few traps. *T. caerulea* captures in Ph and Ph + V-traps were similar. In addition, captures of this predator had a unique pattern; in fact, peaks and minimums alternate regularly if observed per date (Figure 3). This is particularly interesting as the peak dates correspond to 15 days of the life of the pheromone dispenser (counted as the number of days since new dispensers were applied to the traps), while minimum captures were recorded at 30 days of pheromone life (the maximum time it was left in the field). Verbenone significantly decreased catches of “other saproxylic beetles”, with a reduction of 41.28%. However, this difference was almost attributed to *Acanthocinus griseus* (Fabricius), which represents 73.13% of all specimens of this feeding group caught in Ph-traps. No significant difference was observed for non-saproxylic beetles (Table 2).

Considering all feeding groups together, the Ph-traps trapped more than twice the number of specimens than Ph + V-traps (6839 vs. 2601 specimens) (Table 3). However, they mainly trapped bark beetles, which constituted about 87% of the catches. They caught 41 and 42 different beetle species, respectively, with 26 shared species. In addition, the Ph-traps trapped more species of bark beetles (nine vs. six), with four unique species. On the contrary, the Ph + V-traps attracted more non-saproxylic beetle species (seven vs. nine), with five unique species. From Shannon and Simpon’s diversity indices, a higher beetle diversity emerged in Ph + V-traps, and Pielou’s evenness was higher in those traps.

When comparing captures from two of the traps located in the open area (OA) with those of the only two traps located in the forested, covered area (CA), some differences emerged. First, the CA traps caught more than six times as many beetles as the OA traps (Table 3). In addition, even when considering all 10 traps, 61.34% of the catches were collected by the two CA traps. However, most of these catches were bark beetles (87.88%). In fact, the OA traps caught more species in all feeding guilds except for predators, for which the same number of species was found in both environments. In addition, the OA traps caught 13 unique species, while the CA traps only caught 7. The higher number of unique species were distributed in all feeding guilds except for predators. The Shannon diversity index and evenness were higher in the OA traps, while Simpon’s index was lower.

## 4. Discussion

Firstly, we have to be aware that the results of this study almost entirely exclude large beetles that normally are attracted by the *I. sexdentatus* pheromone. As expected, considering the modification of our slot traps (the 6 mm mesh screen applied to the traps), some large beetles abundantly caught at this site (2016–2019) using the *I. sexdentatus* pheromone traps [14] were not collected, or their capture was drastically reduced. Among these species we mention the Melolonthidae *Amadotrogus grassii* (Mainardi), the Elateridae *Lacon punctatus* (Herbst), the two Buprestidae *Chalcophora detrita* (Klug) and *Buprestis novemmaculata* L., and the Cerambycidae *Monochamus galloprovincialis* (Olivier). In addition, some species, such as *T. caerulea* and *T. formicarius*, were probably reduced, as demonstrated by comparing modified traps with non-modified traps [14].

Furthermore, in 2023, the control measures applied to manage the pine pests present in the study area (*M. feytaudi*, *I. sexdentatus*, and *T. destruens*), which included the felling of infested pines, reduced the bark beetle captures compared to the previous study period (2016–2019) in the same area. In fact, considering only the Ph-traps of the present study, the mean numbers per trap of *I. sexdentatus*, *O. erosus*, and *H. ligniperda* in 2023 were 8.6, 573, and 9.1, respectively, while in 2016–2019, considering only modified traps, they were 130, 2237, and 25, respectively [14]. On the contrary, when comparing the captures of the two main predators, this reduction was not observed. In fact, we recorded similar levels of captures of *T. formicarius* (7.3 and 11 specimens/trap/year in 2023 and in 2016–2019, respectively) and an even higher mean number of *T. caerulea* specimens/trap/year in 2023 (48.7) compared with 2016–2019 (28).

Verbenone inhibited bark beetles that can attack weakened living trees, such as *I. sexdentatus* and *O. erosus*, while it attracted *H. ligniperda*. These results on *I. sexdentatus* agree with those of previous studies as the verbenone-induced inhibition of this species has been already described [25,26,27,28]. On the contrary, previous studies on *O. erosus* are inconsistent. For this bark beetle, Paiva et al. [25], in accordance with our results, found an inhibitory effect of verbenone, while Extebeste et al. [29] found an attractive effect when the substance was applied to logs. Furthermore, we found an attractive effect of verbenone for *H. ligniperda*. This result could be consistent with the behavior of this non-aggressive bark beetle, as it is known to exploit aged tissues [30]. Thus, as verbenone is an indicator of poor-quality host tissues [6], it could signal a suitable habitat to *H. ligniperda*. However, our result is not in agreement with previous studies that found no effect of verbenone on this species with the same or lower dosage [26,27].

As regards predators, the effect of verbenone was studied on the *Thanasimus* and *Temnochila* genuses in particular [4]. Predators, such as *T. formicarius* and *T. caerulea*, are proven to be attracted by the *I. sexdentatus* pheromone [14,27,28,29,31,32], which they exploit as a kairomone to locate their prey. Based on previous studies, these two species, in addition to preying on larvae, also prey on bark beetle adults when they colonize new hosts. As a consequence, preferring the first phases of tree colonization, they could be inhibited by verbenone [27,29]. Our study confirms this hypothesis as regards *T. formicarius*; in fact, catches in the Ph + V-traps were lowered by about 73%, similar to Etxebeste and Pajares [27], for which a reduction of 80% was found. However, in contrast with these studies, we found no effect of verbenone on *T. caerula*. In our study, this species seemed particularly sensitive to pheromone release rates as it was the only species for which captures strictly depended on dispenser life (the number of days spent in the trap before renewal). In fact, only higher quantities of pheromone released (most likely during the first two weeks after bait renewal) allowed *T. caerulea* to be caught abundantly.

The other predators, *R. depressus*, *P. otti*, *C. pini*, and *A. ruficorne*, had been already found in bark beetle pheromone traps [31,32,33], showing that they also use these pheromones as kairomones. These species are all predators of bark beetles, particularly of *O. erosus* and *I. sexdentatus* [33,34]. However, no information about how verbenone affects them was previously available. In our study, *R. depressus* was not affected by this substance, while *P. otti*, *C. pini*, and *A. ruficorne* were significantly attracted. This is in agreement with the findings of Lee et al. [2], who stated that predators constitute the prevalent insect-feeding guild in highly degraded wood. Thus, if this type of wood is a better habitat for many predators, verbenone could be a useful signal for this trophic guild. In agreement with this hypothesis, two related species, *Rhizophagus grandis* Gyll. and *Corticeus praetermissus* (Fall), were shown to be attracted to this substance [35].

An inhibitory effect of verbenone on the other saproxylic beetles (considered as a whole) was found. However, this effect was mainly due to only one species, namely *A. griseus*, which, in fact, is a species that attacks weakened or recently dead conifers [36]. The other species are later successional saproxylic beetles; thus, this substance, together with pheromones, is a possible signal of bark beetle feeding activity and aged wood [4,6]; it is likely to be used by them to locate suitable wood. In addition, a higher species diversity and a higher number of unique species of this feeding group were found in traps lured with verbenone, and this is in accordance with the hypothesis of an attractive effect of verbenone on these saproxylic species.

Finally, we found a higher species diversity of saproxylic beetles in open areas than in more forested areas. In fact, in traps located in open areas, there was less dominance of bark beetles and a higher percentage of predators, with a higher total number of species and unique species. This is in agreement with other studies on this topic [37,38] that found higher saproxylic beetle diversity in heterogeneous habitats with a higher level of canopy openness.

## 5. Conclusions

Our study showed that the attractiveness of the *I. sexdentatus* aggregation pheromone for the target species as well as *O. erosus* and *T. formicarius* was reduced by verbenone. This substance could be a signal of intense bark beetle feeding activity and highly degraded wood [4,6]; thus, it could be inhibitory for the two bark bark beetles because they generally prefer weakened living trees. Similarly, *T. formicarius*, also predating on adult bark beetles in the aggregation phase, may prefer lower concentrations of this substance. On the contrary, the less aggressive bark beetle *H. ligniperda*, preferring aged tissues, is attracted as a result. Finally, while two predators, *T. caerulea* and *R. depressus* were not affected, three of the predators analyzed, *P. otti*, *C. pini*, and *A. ruficorne*, were attracted to verbenone as they may use it as a signal for locating a suitable habitat.

## Figures and Tables

**Figure 1 insects-15-00260-f001:**
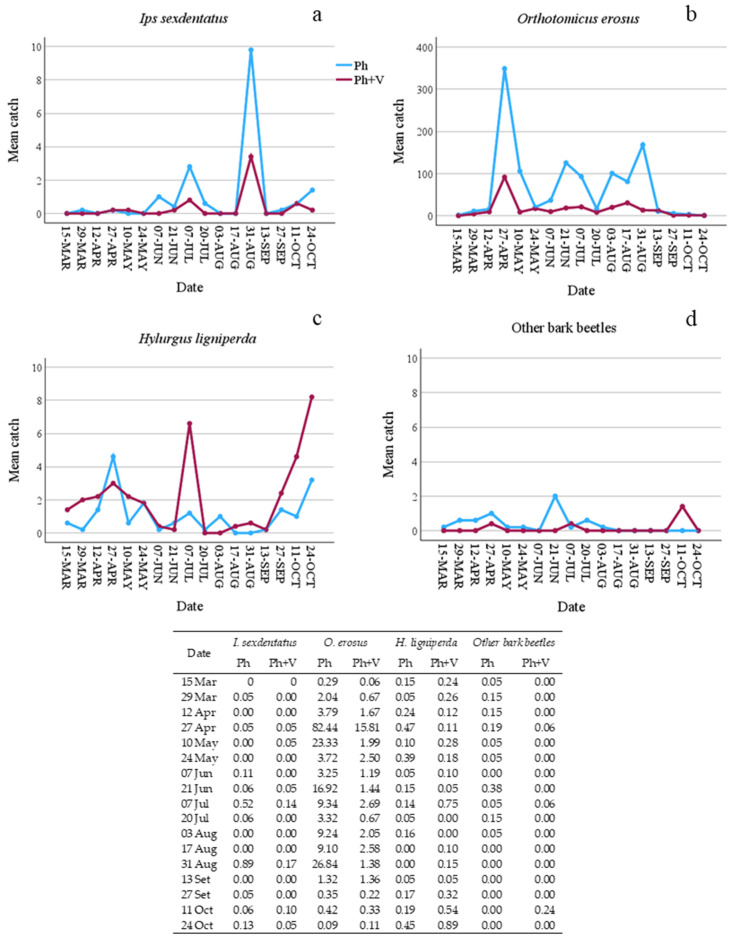
Mean number of bark beetles caught in traps baited with only pheromone (Ph) or with pheromone and verbenone (Ph + V) per sampling date. (**a**) Mean catches of *Ips sexdentatus*; (**b**) mean catches of *Orthotomicus erosus*; (**c**) mean catches of *Hylurgus lingiperda*; (**d**) mean catches of other bark beetle species. Standard errors are shown in table.

**Figure 2 insects-15-00260-f002:**
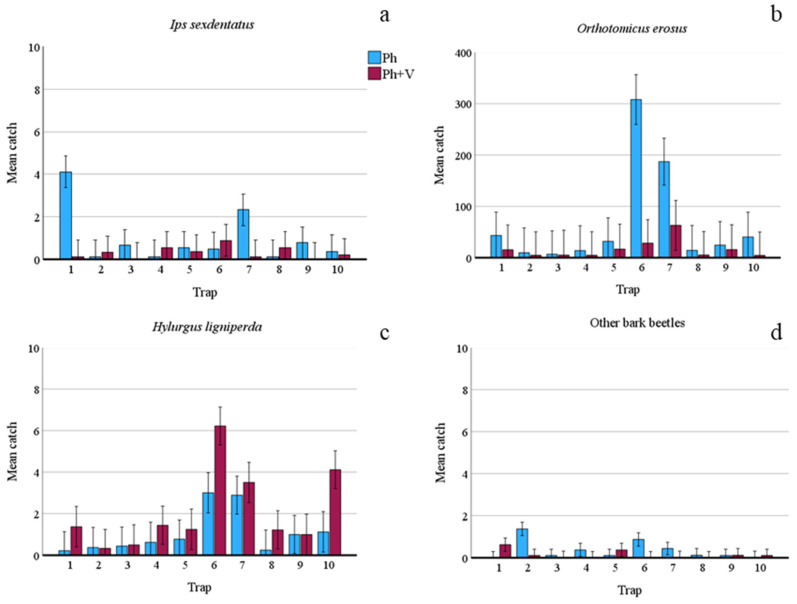
Mean number of bark beetles caught by each slot-trap when baited with only pheromone (Ph) or with pheromone and verbenone (Ph + V) over entire trial period (1 March 2023–24 October 2023). (**a**) Mean catches of *Ips sexdentatus*; (**b**) mean catches of *Orthotomicus erosus*; (**c**) mean catches of *Hylurgus ligniperda*; (**d**) mean catches of other bark beetle species. Bars show standard errors.

**Figure 3 insects-15-00260-f003:**
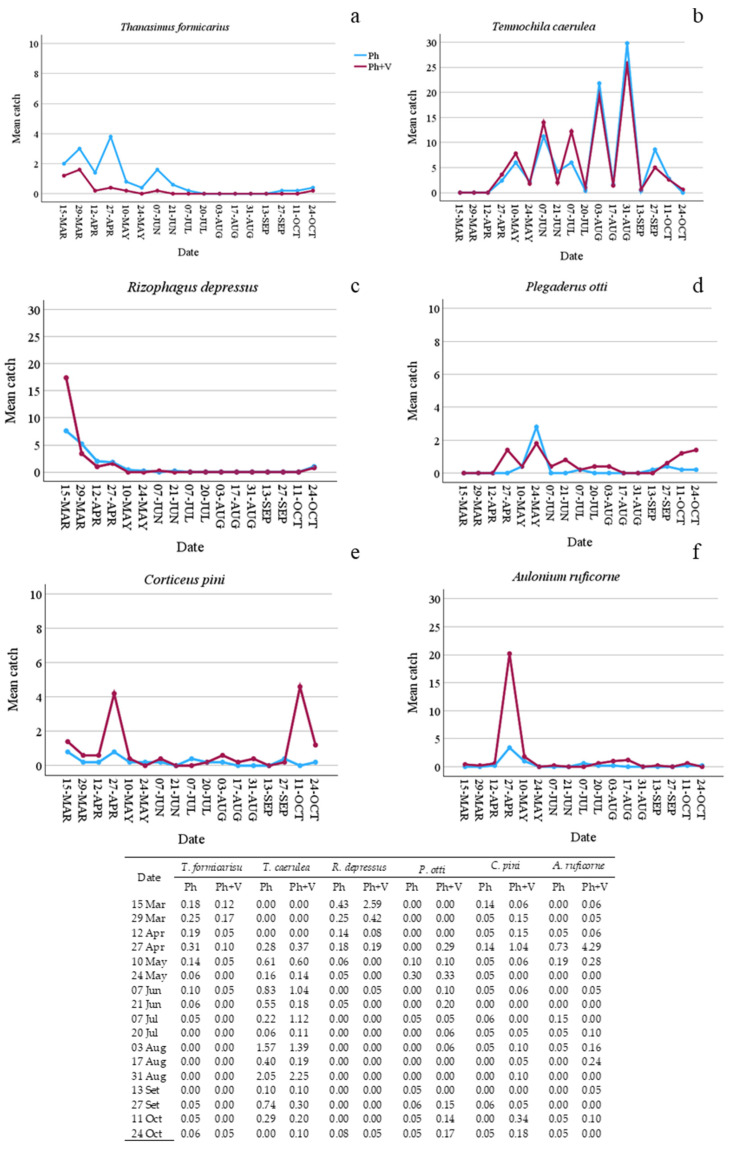
Mean number of beetle predators caught in traps baited with only pheromone (Ph) or with pheromone and verbenone (Ph + V) per sampling date. (**a**) Mean catches of *Thanasimus formicarius*; (**b**) mean catches of *Temnochila caerulea*; (**c**) mean catches of *Rhizophagus depressus*; (**d**) mean catches of *Plegaderus otti*; (**e**) mean catches of *Corticeus pini*; and (**f**) mean catches of *Aulonium ruficorne*. Standard errors are shown in table.

**Figure 4 insects-15-00260-f004:**
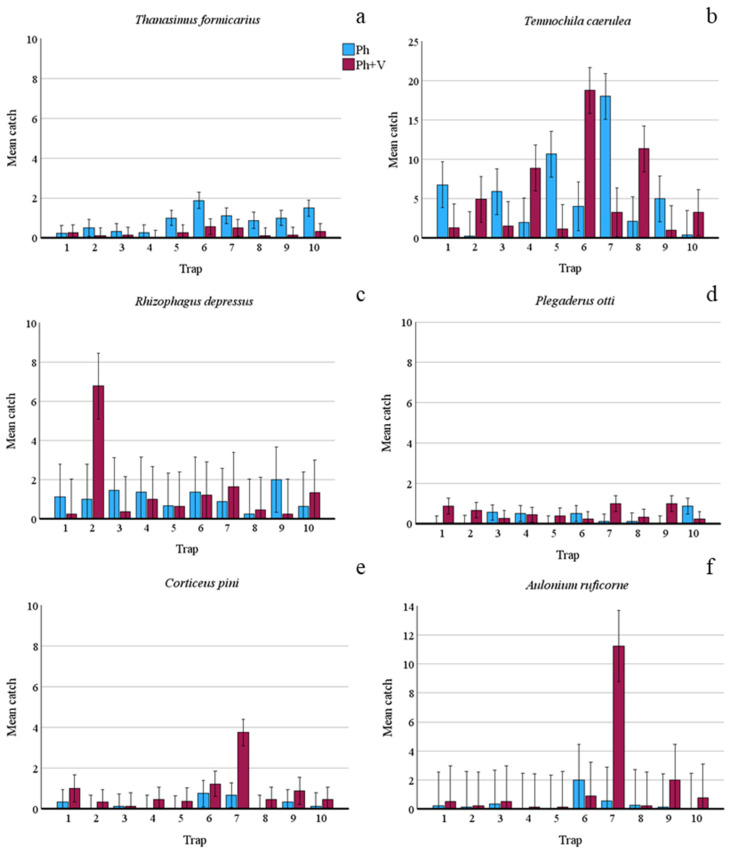
Mean number of beetle predators caught by each slot-trap when baited with only pheromone (Ph) or with pheromone and verbenone (Ph + V) over the whole trial period (1 March 2023–24 October 2023). (**a**) Mean catches of *Thanasimus formicarius*; (**b**) mean catches of *Temnochila caerulea*; (**c**) mean catches of *Rhizophagus depressus*; (**d**) mean catches of *Plegaderus otti*; (**e**) mean catches of *Corticeus pini*; and (**f**) mean catches of *Aulonium ruficorne*. Bars show standard errors.

**Table 1 insects-15-00260-t001:** List of beetle species collected in the pine stand (Grosseto, Italy) during the study period from 1 March 2023 to 24 October 2023 (N.D. = non-identified species).

Species/Feeding Guild	Family	No. of Specimens
Bark Beetles	Curculionidae	7528
*Ips sexdentatus* (Boern)		114
*Crypturgus mediterraneus* Eichhoff		3
*Crypturgus pusillus* (Gyllenhal)		8
*Hylastes linearis* Erichson		1
*Hylurgus ligniperda* (Fabricius)		272
*Hylurgus micklitzi* Wachtl		1
*Orthotomicus erosus* (Wollaston)		7101
*Pityogenes bidentatus* (Herbst)		16
*Xylocleptes biuncus* Reitter		11
*Xyleborus saxesenii* (Ratzeburg)		1
Predators		1628
*Bradycellus verbasci* (Duftschmid)	Carabidae	6
*Dromius meridionalis* Dejean	Carabidae	5
*Olisthopus* Dejean	Carabidae	1
*Thanasimus formicarius* (Linnaeus)	Cleridae	93
*Lacon punctatus* (Herbst)	Elateridae	3
*Paromalus parallelepipedus* (Herbst)	Histeridae	1
*Platysoma elongatum* (Thunberg)	Histeridae	2
*Plegaderus otti* Marseul	Histeridae	67
*Rhizophagus depressus* (Fabricius)	Monotomidae	214
*Temnochila caerulea* (Olivier)	Trogossitidae	976
*Corticeus pini* (Panzer)	Tenebrionidae	95
*Aulonium ruficorne* (Olivier)	Zopheridae	165
Other Saproxylic Beetles		174
N.D.	Anobiidae	27
*Hirticomus hispidus* (Rossi)	Anthicidae	2
N.D.	Anthicidae	1
*Pseudeuparius centromaculatus* (Gyllenhal)	Anthribidae	1
*Anthaxia* Solier	Buprestidae	2
*Buprestis novemmaculata* Linnaeus	Buprestidae	1
*Acanthocinus griseus* (Fabricius)	Cerambycidae	61
*Cerylon* Latreille	Cerylonidae	1
*Brachyderes incanus* (Linnaeus)	Curculionidae	20
*Brachytemnus porcatus* (Germar)	Curculionidae	31
*Carphoborus pini* Eichhoff	Curculionidae	4
*Pissodes castaneus* (De Geer)	Curculionidae	1
*Trinodes hirsutus* (Fabricius)	Dermestidae	1
*Cardiophorus collaris* Erichson	Elateridae	7
*Melanotus* Eschscholtz	Elateridae	1
N.D.	Eucinetidae	2
*Margarinotus* Marseul	Histeridae	1
*Enicmus* Thomson	Latridiidae	2
*Nacerdes melanura* (Linnaeus)	Oedemeridae	1
N.D.	Oedemeridae	1
N.D.	Scraptiidae	2
N.D.	Silvanidae	3
*Diaperis boleti* (Linnaeus)	Tenebrionidae	1
Non-Saproxylic Beetles		110
*Dicladispa testacea* (Linné)	Chrysomelidae	1
*Hydroglyphus pusillus* (Fabricius)	Dytiscidae	1
*Ochthebius* Leach	Hydraenidae	61
*Cryptopleurum* Mulsant	Hydrophilidae	1
*Leiodes* Latreille	Leiodidae	1
*Meligethes* Stephens	Nitidulidae	1
*Pria dulcamarae* (Scopoli)	Nitidulidae	2
*Onthophagus* Latreille	Scarabaeidae	2
*Onthophagus furcatus* (Fabricius)	Scarabaeidae	1
*Pleurophorus mediterranicus* Pittino & Mariani	Scarabaeidae	1
*Isomira* Mulsant	Tenebrionidae	35
*Lagria* Fabricius	Tenebrionidae	3
Total		9440

**Table 2 insects-15-00260-t002:** Beetles caught in pine stand (Grosseto, Italy) by pheromone traps (Ph) and traps baited with pheromone in combination with verbenone (Ph + V). Results of GAMM analysis: coefficients (coeff.) of verbenone effect with respective «F» test of significance values and *p* values; seasonal effect (i.e., smoother effect) with respective estimated degrees of freedom (edf), «F» test of significance values, and *p* values. ns: non-significant.

	Ph-Trap Catches	Ph + V-Trap Catches	Results of GAMM Analysis						
			Intercept	Verbenone Effect			Seasonal Effect		
				Coeff.	F	*p*	Edf	F	*p*
Bark beetles									
*I. sexdentatus*	86	28	−1.237	−0.968	18.37	<0.001	8.187	17.22	<0.001
*O. erosus*	5736	1365	2.680	−1.495	2316.00	<0.001	8.979	531.80	<0.001
*H. ligniperda*	91	181	−0.551	0.643	24.24	<0.001	8.033	17.77	<0.001
Predators									
*T. formicarius*	73	20	−1.098	−1.323	26.93	<0.001	4.04	17.53	<0.001
*T. caerulea*	487	489	0.662	0.004	0.00	ns	7.478	42.53	<0.001
*R. depressus*	92	122	−2.580	0.076	0.24	ns	4.537	59.94	<0.001
*P. otti*	22	45	−2.049	0.716	7.48	<0.01	5.41	7.41	<0.001
*C. pini*	20	75	−2.238	1.343	27.54	<0.001	7.985	7.82	<0.001
*A. ruficorne*	30	135	−2.978	1.214	30.82	<0.001	8.453	36.52	<0.001
Other saproxylic beetles									
total	110	64	0.125	−0.524	10.92	<0.001	1.555	5.17	ns
Non-saproxylic beetles									
total	50	60	−1.280	0.199	1.08	ns	6.995	18.68	<0.001

**Table 3 insects-15-00260-t003:** Diversity indices for each feeding guild. Catches were sorted by substance (pheromone—h; pheromone + verbenone—Ph + V) and environment (open area—OA; covered area—CA).

	Species No.		Unique Species		Relative Abundance (%)		Species No.		Unique Species		Relative Abundance (%)	
	Ph	Ph + V	Ph	Ph + V	Ph	Ph + V	OA	CA	OA	CA	OA	CA
Bark beetles	9	6	4	1	86.91	60.94	6	4	3	1	64.69	87.88
Predators	10	11	1	2	10.76	34.29	10	10	5	3	29.53	11.33
Other saproxylic beetles	15	16	9	8	1.61	2.46	10	8	2	2	3.37	0.60
Non-saproxylic beetles	7	9	3	5	0.73	2.31	6	4	3	1	2.44	0.19
Total	41	42	15	16			32	26	13	7		
Trap no.	10	10					2	2				
Total specimens	6839	2601					861	5791				
Shared species	26						19					
Simpson’s index	0.709	0.322					0.378	0.725				
Shannon index	0.80	1.70					1.61	0.62				
Evenness	0.149	0.313					0.322	0.132				

## Data Availability

The data presented in this study are available on request from the corresponding author.

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
