# Peer review of "Verbenone Affects the Behavior of Insect Predators and Other Saproxylic Beetles Differently: Trials Using Pheromone-Baited Bark Beetle Traps"

_insects, 2024, doi:10.3390/insects15040260_

Round 1
Reviewer 1 Report
Comments and Suggestions for Authors
Dear authors,
This manuscript focuses on the effects of verbenone on saproxylic beetles in a Mediterranean pine forest in Tuscany (Italy). The study is well-conceived and offers valuable insights into the impact of verbenone, a widely studied bark beetle aggregation inhibitor, on insect populations.
The experimental design, using verbenone pouches applied to Ips sexdentatus pheromone traps, is innovative and provides a robust framework for comparing beetle captures. The comprehensive collection of data over the spring-summer period of 2023, with catches recorded every two weeks, adds depth and reliability to the findings.
The results are particularly interesting, showcasing a diverse range of beetle species across 32 families and 57 species. The significant reduction in captures for certain beetles, such as I. sexdentatus and O. erosus, and the increase in captures for others in the presence of verbenone highlight the complexity of interactions within the ecosystem. Furthermore, the discovery of previously unstudied predator species in relation to verbenone effects enriches the scientific understanding of this area.
Overall, this manuscript presents valuable findings and contributes meaningfully to the existing body of knowledge on bark beetle ecology and management. I recommend the manuscript for publication after minor revisions for clarity and coherence.
I have only minor requests:
1) Line 8, add Italy after ‘Tuscan pine forest, change in “the impact of verbenone”,
2) Line 21, write 9,440 beetles
3) Line 27, i.e. in italics, no i.e..
4) The fist science in the Introduction (line 36-37) could benefit from some restructuring for improved readability. Avoid starting with ‘In is known that’, as it is redundant and can be removed to make the sentence more direct. It could something like: In forest ecology, it is well understood the assembly of saproxylic beetles varies depending on the condition of the wood. Initially, more demanding bark beetle species colonize the tree, followed by later saproxylic species in the subsequent stages of wood decay.
5) Change ‘to complete in ‘to achieve’
6) Line 86 add: the maritime pine bast scale, Matsucoccus feytaudi (Ducasse)
7) Line 94 incorporate quotation marks in 'Super Forest'
8) Line 101, add the name of the commercial product after: Traps were baited with a commercial blend of the I. sexdentatus aggregation pheromone,XXX (Serbios S.r.l.). This pheromone is known to be also effective… and comprise comprising ipsenol, ipsdienol, and 2-methyl-3-buten-2-ol as the main components. No pesticides were used to treat the collection containers.
9) Line 117, add: like the buprestid beetle, Chalcophora detrita detrita. By the way, this is a non-native species, are you sure it is classified as endangered species?
10) Line 123 Add a space 40 mg/24h
11) Line 125. What do you mean by “wetted pad”?
12) Line 180 and further on very time 1000 numbers are reported. 9,440
13) Line 204 Specify the study period in the caption of Tab. 1
14) Line 227 It should be, Mean catches per trap and per control date of bark beetles collected in the 10 slot traps. If so, please add also SD or SR. If it is too confusing the figure, provide a supplementary tab. with these data. Please add statistics to the fig.
15) It is difficult to understand which information provide fig. 2. The 10 numbered traps were alternately with and without verbenone. So, for instance in trap 1 you are comparing the mean biweekly catches with (red) and without (blue) verbenone all over the 17 periods. If it is like so please specify better in the caption (231) also of fig. 3.
16) Fig. 3 and 4 should be switched , following the same order of Fig. and Fig. 2.
Comments on the Quality of English LanguageThe English quality in this manuscript is acceptable; however, it would gain from a review by a native speaker proficient in scientific writing.
Author Response
Response to the referees
Referee 1
- Line 8, add Italy after ‘Tuscan pine forest, change in “the impact of verbenone”,
We followed the suggestions and we moved the reference to the study area at the end of the sentence.
- Line 21, write 9,440 beetles
Done
- Line 27, i.e. in italics, no i.e..
Done
- The fist science in the Introduction (line 36-37) could benefit from some restructuring for improved readability. Avoid starting with ‘In is known that’, as it is redundant and can be removed to make the sentence more direct. It could something like: In forest ecology, it is well understood the assembly of saproxylic beetles varies depending on the condition of the wood. Initially, more demanding bark beetle species colonize the tree, followed by later saproxylic species in the subsequent stages of wood decay.
We rephrased the sentence accordingly, following the referee’s proposal.
- Change ‘to complete in ‘to achieve’
We complied with referee’s suggestion. No mention of the line was given but we figured it was most probably before reference [5], line 44 of the revised manuscript.
- Line 86 add: the maritime pine bast scale, Matsucoccus feytaudi (Ducasse).
Done
- Line 94 incorporate quotation marks in 'Super Forest'
Done
- Line 101, add the name of the commercial product after: Traps were baited with a commercial blend of the I. sexdentatus aggregation pheromone,XXX (Serbios S.r.l.). This pheromone is known to be also effective… and comprise comprising ipsenol, ipsdienol, and 2-methyl-3-buten-2-ol as the main components. No pesticides were used to treat the collection containers.
Done
- Line 117, add: like the buprestid beetle, Chalcophora detrita detrita. By the way, this is a non-native species, are you sure it is classified as endangered species?
Yes, C. detrita is listed as endangered in Italy. Its introduction to Italy dates back at least a millennium; that could be a reason why its conservation is legislated in Tuscany as well as nationally. (https://www.iucn.it/scheda-2018.php?id=1745538227) (https://www.regione.toscana.it/documents/10180/392141/dcae9a1c7cf71b27283967bc0ab73be4_lr56del6aprile2000sir.pdf/8fc35c60-b0f0-4b6d-9009-96c7511871c1)
- Line 123 Add a space 40 mg/24h
Done
- Line 125. What do you mean by “wetted pad”?
A square of folded absorbent paper wetted with verbenone. The folding of a 90 sq. cm piece of abs. paper gives it a pad thickness.
- Line 180 and further on very time 1000 numbers are reported. 9,440
Done
- Line 204 Specify the study period in the caption of Tab. 1
Done
- Line 227 It should be, Mean catches per trap and per control date of bark beetles collected in the 10 slot traps. If so, please add also SD or SR. If it is too confusing the figure, provide a supplementary tab. with these data. Please add statistics to the fig.
We added a supplementary table with standard errors as suggested. However, since related statistics are available in table 2, we prefer not to alter the figure any further as this kind of chart does not support the display of more statistical significance.
- It is difficult to understand which information provide fig. 2. The 10 numbered traps were alternately with and without verbenone. So, for instance in trap 1 you are comparing the mean biweekly catches with (red) and without (blue) verbenone all over the 17 periods. If it is like so please specify better in the caption (231) also of fig. 3.
The referee is correct. That is the information provided by the figure. We rephrased the caption to make it clearer.
- 3 and 4 should be switched , following the same order of Fig. and Fig. 2.
Done
Response to the referees
Referee 1
- Line 8, add Italy after ‘Tuscan pine forest, change in “the impact of verbenone”,
We followed the suggestions and we moved the reference to the study area at the end of the sentence.
- Line 21, write 9,440 beetles
Done
- Line 27, i.e. in italics, no i.e..
Done
- The fist science in the Introduction (line 36-37) could benefit from some restructuring for improved readability. Avoid starting with ‘In is known that’, as it is redundant and can be removed to make the sentence more direct. It could something like: In forest ecology, it is well understood the assembly of saproxylic beetles varies depending on the condition of the wood. Initially, more demanding bark beetle species colonize the tree, followed by later saproxylic species in the subsequent stages of wood decay.
We rephrased the sentence accordingly, following the referee’s proposal.
- Change ‘to complete in ‘to achieve’
We complied with referee’s suggestion. No mention of the line was given but we figured it was most probably before reference [5], line 44 of the revised manuscript.
- Line 86 add: the maritime pine bast scale, Matsucoccus feytaudi (Ducasse).
Done
- Line 94 incorporate quotation marks in 'Super Forest'
Done
- Line 101, add the name of the commercial product after: Traps were baited with a commercial blend of the I. sexdentatus aggregation pheromone,XXX (Serbios S.r.l.). This pheromone is known to be also effective… and comprise comprising ipsenol, ipsdienol, and 2-methyl-3-buten-2-ol as the main components. No pesticides were used to treat the collection containers.
Done
- Line 117, add: like the buprestid beetle, Chalcophora detrita detrita. By the way, this is a non-native species, are you sure it is classified as endangered species?
Yes, C. detrita is listed as endangered in Italy. Its introduction to Italy dates back at least a millennium; that could be a reason why its conservation is legislated in Tuscany as well as nationally. (https://www.iucn.it/scheda-2018.php?id=1745538227) (https://www.regione.toscana.it/documents/10180/392141/dcae9a1c7cf71b27283967bc0ab73be4_lr56del6aprile2000sir.pdf/8fc35c60-b0f0-4b6d-9009-96c7511871c1)
- Line 123 Add a space 40 mg/24h
Done
- Line 125. What do you mean by “wetted pad”?
A square of folded absorbent paper wetted with verbenone. The folding of a 90 sq. cm piece of abs. paper gives it a pad thickness.
- Line 180 and further on very time 1000 numbers are reported. 9,440
Done
- Line 204 Specify the study period in the caption of Tab. 1
Done
- Line 227 It should be, Mean catches per trap and per control date of bark beetles collected in the 10 slot traps. If so, please add also SD or SR. If it is too confusing the figure, provide a supplementary tab. with these data. Please add statistics to the fig.
We added a supplementary table with standard errors as suggested. However, since related statistics are available in table 2, we prefer not to alter the figure any further as this kind of chart does not support the display of more statistical significance.
- It is difficult to understand which information provide fig. 2. The 10 numbered traps were alternately with and without verbenone. So, for instance in trap 1 you are comparing the mean biweekly catches with (red) and without (blue) verbenone all over the 17 periods. If it is like so please specify better in the caption (231) also of fig. 3.
The referee is correct. That is the information provided by the figure. We rephrased the caption to make it clearer.
- 3 and 4 should be switched , following the same order of Fig. and Fig. 2.
Done
Response to the referees
Referee 1
- Line 8, add Italy after ‘Tuscan pine forest, change in “the impact of verbenone”,
We followed the suggestions and we moved the reference to the study area at the end of the sentence.
- Line 21, write 9,440 beetles
Done
- Line 27, i.e. in italics, no i.e..
Done
- The fist science in the Introduction (line 36-37) could benefit from some restructuring for improved readability. Avoid starting with ‘In is known that’, as it is redundant and can be removed to make the sentence more direct. It could something like: In forest ecology, it is well understood the assembly of saproxylic beetles varies depending on the condition of the wood. Initially, more demanding bark beetle species colonize the tree, followed by later saproxylic species in the subsequent stages of wood decay.
We rephrased the sentence accordingly, following the referee’s proposal.
- Change ‘to complete in ‘to achieve’
We complied with referee’s suggestion. No mention of the line was given but we figured it was most probably before reference [5], line 44 of the revised manuscript.
- Line 86 add: the maritime pine bast scale, Matsucoccus feytaudi (Ducasse).
Done
- Line 94 incorporate quotation marks in 'Super Forest'
Done
- Line 101, add the name of the commercial product after: Traps were baited with a commercial blend of the I. sexdentatus aggregation pheromone,XXX (Serbios S.r.l.). This pheromone is known to be also effective… and comprise comprising ipsenol, ipsdienol, and 2-methyl-3-buten-2-ol as the main components. No pesticides were used to treat the collection containers.
Done
- Line 117, add: like the buprestid beetle, Chalcophora detrita detrita. By the way, this is a non-native species, are you sure it is classified as endangered species?
Yes, C. detrita is listed as endangered in Italy. Its introduction to Italy dates back at least a millennium; that could be a reason why its conservation is legislated in Tuscany as well as nationally. (https://www.iucn.it/scheda-2018.php?id=1745538227) (https://www.regione.toscana.it/documents/10180/392141/dcae9a1c7cf71b27283967bc0ab73be4_lr56del6aprile2000sir.pdf/8fc35c60-b0f0-4b6d-9009-96c7511871c1)
- Line 123 Add a space 40 mg/24h
Done
- Line 125. What do you mean by “wetted pad”?
A square of folded absorbent paper wetted with verbenone. The folding of a 90 sq. cm piece of abs. paper gives it a pad thickness.
- Line 180 and further on very time 1000 numbers are reported. 9,440
Done
- Line 204 Specify the study period in the caption of Tab. 1
Done
- Line 227 It should be, Mean catches per trap and per control date of bark beetles collected in the 10 slot traps. If so, please add also SD or SR. If it is too confusing the figure, provide a supplementary tab. with these data. Please add statistics to the fig.
We added a supplementary table with standard errors as suggested. However, since related statistics are available in table 2, we prefer not to alter the figure any further as this kind of chart does not support the display of more statistical significance.
- It is difficult to understand which information provide fig. 2. The 10 numbered traps were alternately with and without verbenone. So, for instance in trap 1 you are comparing the mean biweekly catches with (red) and without (blue) verbenone all over the 17 periods. If it is like so please specify better in the caption (231) also of fig. 3.
The referee is correct. That is the information provided by the figure. We rephrased the caption to make it clearer.
- 3 and 4 should be switched , following the same order of Fig. and Fig. 2.
Done
Response to the referees
Referee 1
- Line 8, add Italy after ‘Tuscan pine forest, change in “the impact of verbenone”,
We followed the suggestions and we moved the reference to the study area at the end of the sentence.
- Line 21, write 9,440 beetles
Done
- Line 27, i.e. in italics, no i.e..
Done
- The fist science in the Introduction (line 36-37) could benefit from some restructuring for improved readability. Avoid starting with ‘In is known that’, as it is redundant and can be removed to make the sentence more direct. It could something like: In forest ecology, it is well understood the assembly of saproxylic beetles varies depending on the condition of the wood. Initially, more demanding bark beetle species colonize the tree, followed by later saproxylic species in the subsequent stages of wood decay.
We rephrased the sentence accordingly, following the referee’s proposal.
- Change ‘to complete in ‘to achieve’
We complied with referee’s suggestion. No mention of the line was given but we figured it was most probably before reference [5], line 44 of the revised manuscript.
- Line 86 add: the maritime pine bast scale, Matsucoccus feytaudi (Ducasse).
Done
- Line 94 incorporate quotation marks in 'Super Forest'
Done
- Line 101, add the name of the commercial product after: Traps were baited with a commercial blend of the I. sexdentatus aggregation pheromone,XXX (Serbios S.r.l.). This pheromone is known to be also effective… and comprise comprising ipsenol, ipsdienol, and 2-methyl-3-buten-2-ol as the main components. No pesticides were used to treat the collection containers.
Done
- Line 117, add: like the buprestid beetle, Chalcophora detrita detrita. By the way, this is a non-native species, are you sure it is classified as endangered species?
Yes, C. detrita is listed as endangered in Italy. Its introduction to Italy dates back at least a millennium; that could be a reason why its conservation is legislated in Tuscany as well as nationally. (https://www.iucn.it/scheda-2018.php?id=1745538227) (https://www.regione.toscana.it/documents/10180/392141/dcae9a1c7cf71b27283967bc0ab73be4_lr56del6aprile2000sir.pdf/8fc35c60-b0f0-4b6d-9009-96c7511871c1)
- Line 123 Add a space 40 mg/24h
Done
- Line 125. What do you mean by “wetted pad”?
A square of folded absorbent paper wetted with verbenone. The folding of a 90 sq. cm piece of abs. paper gives it a pad thickness.
- Line 180 and further on very time 1000 numbers are reported. 9,440
Done
- Line 204 Specify the study period in the caption of Tab. 1
Done
- Line 227 It should be, Mean catches per trap and per control date of bark beetles collected in the 10 slot traps. If so, please add also SD or SR. If it is too confusing the figure, provide a supplementary tab. with these data. Please add statistics to the fig.
We added a supplementary table with standard errors as suggested. However, since related statistics are available in table 2, we prefer not to alter the figure any further as this kind of chart does not support the display of more statistical significance.
- It is difficult to understand which information provide fig. 2. The 10 numbered traps were alternately with and without verbenone. So, for instance in trap 1 you are comparing the mean biweekly catches with (red) and without (blue) verbenone all over the 17 periods. If it is like so please specify better in the caption (231) also of fig. 3.
The referee is correct. That is the information provided by the figure. We rephrased the caption to make it clearer.
- 3 and 4 should be switched , following the same order of Fig. and Fig. 2.
Done
Response to the referees
Referee 1
- Line 8, add Italy after ‘Tuscan pine forest, change in “the impact of verbenone”,
We followed the suggestions and we moved the reference to the study area at the end of the sentence.
- Line 21, write 9,440 beetles
Done
- Line 27, i.e. in italics, no i.e..
Done
- The fist science in the Introduction (line 36-37) could benefit from some restructuring for improved readability. Avoid starting with ‘In is known that’, as it is redundant and can be removed to make the sentence more direct. It could something like: In forest ecology, it is well understood the assembly of saproxylic beetles varies depending on the condition of the wood. Initially, more demanding bark beetle species colonize the tree, followed by later saproxylic species in the subsequent stages of wood decay.
We rephrased the sentence accordingly, following the referee’s proposal.
- Change ‘to complete in ‘to achieve’
We complied with referee’s suggestion. No mention of the line was given but we figured it was most probably before reference [5], line 44 of the revised manuscript.
- Line 86 add: the maritime pine bast scale, Matsucoccus feytaudi (Ducasse).
Done
- Line 94 incorporate quotation marks in 'Super Forest'
Done
- Line 101, add the name of the commercial product after: Traps were baited with a commercial blend of the I. sexdentatus aggregation pheromone,XXX (Serbios S.r.l.). This pheromone is known to be also effective… and comprise comprising ipsenol, ipsdienol, and 2-methyl-3-buten-2-ol as the main components. No pesticides were used to treat the collection containers.
Done
- Line 117, add: like the buprestid beetle, Chalcophora detrita detrita. By the way, this is a non-native species, are you sure it is classified as endangered species?
Yes, C. detrita is listed as endangered in Italy. Its introduction to Italy dates back at least a millennium; that could be a reason why its conservation is legislated in Tuscany as well as nationally. (https://www.iucn.it/scheda-2018.php?id=1745538227) (https://www.regione.toscana.it/documents/10180/392141/dcae9a1c7cf71b27283967bc0ab73be4_lr56del6aprile2000sir.pdf/8fc35c60-b0f0-4b6d-9009-96c7511871c1)
- Line 123 Add a space 40 mg/24h
Done
- Line 125. What do you mean by “wetted pad”?
A square of folded absorbent paper wetted with verbenone. The folding of a 90 sq. cm piece of abs. paper gives it a pad thickness.
- Line 180 and further on very time 1000 numbers are reported. 9,440
Done
- Line 204 Specify the study period in the caption of Tab. 1
Done
- Line 227 It should be, Mean catches per trap and per control date of bark beetles collected in the 10 slot traps. If so, please add also SD or SR. If it is too confusing the figure, provide a supplementary tab. with these data. Please add statistics to the fig.
We added a supplementary table with standard errors as suggested. However, since related statistics are available in table 2, we prefer not to alter the figure any further as this kind of chart does not support the display of more statistical significance.
- It is difficult to understand which information provide fig. 2. The 10 numbered traps were alternately with and without verbenone. So, for instance in trap 1 you are comparing the mean biweekly catches with (red) and without (blue) verbenone all over the 17 periods. If it is like so please specify better in the caption (231) also of fig. 3.
The referee is correct. That is the information provided by the figure. We rephrased the caption to make it clearer.
- 3 and 4 should be switched , following the same order of Fig. and Fig. 2.
Done
Please see the attached file

Reviewer 2 Report
Comments and Suggestions for Authors
The proposed research is interesting and provides useful data and new knowledge.
Please, adapt the title to the contents of the work.
The summary is rather complicated.
In general, in the text make some references to plant-insect relationships.
The graphs must be modified in the form of histograms because the subsequent data are not in relation one-one.
If possible, make some comments on the different presence of the two sexes (male and female). Other annotations are highlighted in the attached text.

Comments on the Quality of English LanguageModerate
